



# Biogeochemical evolution of ponded meltwater in a High Arctic subglacial tunnel

Ashley J. Dubnick[1,2], Rachel L. Spietz[3], Brad D. Danielson[4], Mark L. Skidmore[1], Eric S. Boyd[3], Dave Burgess[4], Charvanaa Dhoonmoon[2], Martin Sharp[2].

[1]Department of Earth Science, Montana State University, Bozeman, Montana, 59717, USA
[2]Department of Earth and Atmospheric Sciences, University of Alberta, Edmonton, Alberta, T6G 2E3, Canada
[3]Department of Microbiology and Cell Biology, Montana State University, Bozeman, MT, 59717, USA
[4]Geological Survey of Canada, 601 Booth Street, Ottawa, Ontario, K1A 0E8, Canada

*Correspondence to*: Ashley J. Dubnick (ashley.dubnick@montana.edu)

**Abstract.** Subglacial environments comprise ~10% of Earth's land surface, host active microbial ecosystems, and are important components of global biogeochemical cycles. However, the broadly inaccessible nature of subglacial systems has left them vastly understudied, and research to date has been limited to laboratory experiments or field measurements using basal ice or subglacial water accessed through boreholes or from the glacier margin. In this study, we extend our understanding of subglacial biogeochemistry and microbiology to include observations of a slushy pond of water that occupied a remnant meltwater channel beneath a polythermal glacier in the Canadian High Arctic over winter. The hydraulics and geochemistry of the system suggest that the pond water originated as late-season, ice-marginal runoff with less than ~15% solute contribution from subglacial sources. Over the eight months of persistent sub-zero regional temperatures, the pond gradually froze, cryo-concentrating solutes in the residual water by up to seven times. Despite cryo-concentration and the likely influx of some subglacial solute, the pond was depleted in only the most labile and biogeochemically-relevant compounds, including ammonium, phosphate, and dissolved organic matter, including a potentially labile tyrosine-like component. DNA amplicon sequencing revealed decreasing microbial diversity with distance into the meltwater channel. The pond at the terminus of the channel hosted a microbial community inherited from late-season meltwater, which was dominated by only six taxa related to known psychrophilic/psychrotolerant heterotrophs that have high metabolic diversity and broad habitat ranges. Collectively, our findings suggest that generalist microbes from the extraglacial or supraglacial environments can become established in subglacial aquatic systems and deplete reservoirs of nutrients and dissolved organic carbon over a period of months. These findings extend our understanding of the microbial and biogeochemical evolution of subglacial aquatic ecosystems and the extent of their habitability.



## 1 Introduction

Subglacial environments currently cover approximately 10% of Earth's land surface and have occupied most of these areas for thousands of years. The broadly inaccessible nature of these systems has left them vastly understudied relative to other terrestrial ecosystems. Yet, over the last two decades efforts to explore subglacial sediment, water, and ice have consistently characterized them as active microbial and chemical systems that are important to global-scale biogeochemical cycles (e.g. Boyd *et al.*, 2010; Hawkings *et al.*, 2014; Wadham *et al.*, 2019; Kellerman *et al.*, 2021).

Microorganisms are ubiquitous in subglacial systems. They are found in subglacial water at concentrations ranging from 1 x $10^2$ to 1 x $10^5$ cells ml$^{-1}$ (Christner et al., 2014; Sheridan et al., 2003; Grasby et al., 2003) and are even more abundant in subglacial sediment at concentrations of $10^6$ to $10^7$ cells g$^{-1}$ of sediment (Sharp et al., 1999; Lanoil et al., 2009). Studies consistently find that among these cells exists a metabolically-active microbial community that is often phylogenetically and functionally diverse (Vick-Majors et al., 2016; Yde et al., 2010; Christner et al., 2014; Hamilton et al., 2013). Some of these

subglacial ecosystems are sustained by reservoirs of organic matter (Hood et al., 2015) acquired as glaciers override soil and vegetation (Christ et al., 2021), marine deposits (Wadham et al., 2012; Michaud et al., 2016), and organic-rich shales (Wadham et al., 2004; Grasby et al., 2003), or from the in-wash of supraglacial or ice-marginal organic matter (Tranter et al., 2005; Andrews et al., 2018). This organic matter can support microorganisms with heterotrophic metabolisms, even under anaerobic conditions involving methanogenesis as a terminal process (Boyd et al., 2010; Stibal et al., 2012a; Wadham et al., 2012).

Primary production in many subglacial ecosystems is dominated by lithotrophic microbial metabolisms (Kayani et al., 2018; Boyd et al., 2014; Christner et al., 2014; Dunham et al., 2021), in which organisms utilize solutes or elements from the underlying bedrock to liberate chemical energy. Microbially-mediated redox reactions in subglacial systems occur via the oxidation of reduced inorganic compounds such as ferrous iron ($Fe^{2+}$), sulfide ($S^{2-}$), hydrogen ($H_2$), ammonium ($NH_4^+$), and methane ($CH_4$,) coupled to the reduction of species such as oxygen ($O_2$), nitrate ($NO_3^-$), carbon dioxide ($CO_2$), or sulfate ($SO_4^{2-}$

) (Miteva et al., 2004; Boyd et al., 2011; Yde et al., 2010; Stibal et al., 2012a; Wadham et al., 2004; Tranter et al., 2002; Boyd et al., 2014; Dunham et al., 2021; Achberger et al., 2016; Christner et al., 2014; Michaud et al., 2017). Glacial comminution serves to liberate, and in some cases even produce (Telling et al., 2015; Macdonald et al., 2018; Gill-Olivas et al., 2021), redox-sensitive species for subglacial ecosystems. Subglacial bedrock and freshly comminuted sediments can also contain high concentrations of labile nutrients required to support microbial activity, including phosphorus (Föllmi et al., 2009; Hawkings

et al., 2016; Hodson et al., 2004), iron (Bhatia et al., 2013; Hawkings et al., 2014; Schroth et al., 2014), and nitrogen (Hodson et al., 2005; Lawson et al., 2014; Wadham et al., 2016).

The research that has informed our understanding of the microbiology and geochemistry of subglacial systems has been conducted using samples of basal ice (e.g. Barker *et al.*, 2010; Montross *et al.*, 2014; Dubnick *et al.*, 2020) and subglacial water accessed via boreholes (e.g. Tranter, 2003; Christner *et al.*, 2014) or from the glacier margin (e.g. Foght *et al.*, 2004;

Skidmore *et al.*, 2005; Boyd et al., 2011; Sheik *et al.*, 2015). Though these environmental samples have advanced our understanding of subglacial biogeochemistry, the inaccessible and dynamic nature of subglacial environments make it difficult





to interpret observations in the context of residence times, water histories, or solute sources, and thus fully resolve the processes that occur in-situ or the rates at which they occur. In this study, we extend our understanding of subglacial biogeochemistry and microbiology to include observations on the evolution of meltwater after it overwinters at the endpoint of a remnant

subglacial channel that extends 467 m beneath a glacier. The nature of this system provides us with a unique opportunity to observe the biogeochemical signatures of a relatively isolated subglacial system with strong constraints on the potential solute sources, water histories, and subglacial residence times.

We compared the biogeochemistry and microbiology of water samples from a pond at the terminus of the channel to their water and solute sources: late season runoff (represented by ice samples collected from frozen sections of the channel floor)

and basal solute (represented by basal ice exposed along the tunnel walls). We used water isotopes ($\delta^2$H-$\delta^{18}$O) and a conservative geochemical tracer (Cl$^-$) to quantify the extent to which the late-season water froze and evaporated *in situ*. Using these results, a geochemical freeze-fractionation model was developed to 1) identify solutes that appear to behave conservatively and those in which there is an apparent in-situ source or sink, 2) evaluate the extent to which basal solutes may have contributed to the waterbody, and 3) in combination with 16S rRNA gene amplicon sequencing, explore evidence for *in*

*situ* microbial activity and nutrient cycling. Collectively, the biogeochemical and microbial datasets provide evidence of *in situ* microbial activity and show that a distinct microbial community can develop in a subglacial waterbody and deplete reservoirs of the most labile nutrients within months.

## 2    Methods

### 2.1 Field Site

The Sverdrup Glacier is a 25-km long tidewater glacier that overrides metasedimentary rocks and gneiss bedrock (Harrison et al., 2016) while draining a ~800 km$^2$ northwest sector of the Devon Ice Cap, Canadian Arctic (Figure 1). Ice velocities along the Sverdrup Glacier are moderate, ranging from ~30 m yr$^{-1}$ near the glacier headwall to ~75 m yr$^{-1}$ near the terminus (Van Wychen et al., 2017; Cress and Wyness, 1961). While some areas of the glacier are probably frozen to the bed (Van Wychen et al., 2017), flow rates along much of the glacier are influenced by basal sliding or enhanced deformation of basal ice,

suggesting that basal ice temperatures, in at least some areas of the bed, approach the pressure-melting point (Van Wychen et al., 2017; Burgess et al., 2005). A two-fold increase in surface velocity of the Sverdrup Glacier first measured during the summer of 1961 (Cress and Wyness, 1961) suggests meltwater reaches the bed and significantly reduces friction.

Consistent with other observations across the Canadian high Arctic, net mass balance of the Sverdrup glacier basin is controlled almost entirely by surface melt (Koerner, 2005). Surface temperatures measured on the Sverdrup Glacier from 2005-2021

indicate a daily average of 4.5°C over the summer melt season, which extends from early June to late August. Most of the melt generated on the Sverdrup Glacier drains ice-marginally, with the remainder draining via supraglacial streams that reach the bed through moulins near the marine-terminating glacier snout (Keeler, 1964; Koerner, 1961). Winter temperatures average -26°C, suggesting that penetration of the winter cold wave likely leaves ice frozen to the bed along the lateral margins, as is the







Figure 1: (a) Sverdrup glacier including radar and laser altimeter transects (Transect A-B; Paden et al., 2019) and subglacial tunnel survey. Base map is the Hillshade image from ArcticDEM (Porter et al., 2018), (b) Devon Ice Cap (ESRI satellite base map), showing the location of the Sverdrup Glacier. (c) cross sectional profile showing the bedrock and ice surface along A-B (Paden et al., 2019) with the approximate distance of the subglacial pond from the lateral margin imposed on this transect.



case for many polythermal glaciers (Bingham et al., 2006; Irvine-Fynn et al., 2011). The area explored in this study focuses
on ponded water located at the endpoint of a remnant subglacial tunnel along the eastern margin, ~3 km from the glacier
terminus (Figure 1).

**2.2 Field Methods**

We visited the Sverdrup subglacial tunnel in May 2019 and mapped the system using a TruPulse®360 (Laser Tech (LTI),
Colorado USA) rangefinder relative to a known reference point at the glacier margin for which coordinates were collected
using a handheld GPS. The rangefinder device was mounted on a tripod with a known height, and the azimuth, inclination,
horizontal distance, and vertical distance were collected for a back-sight and fore-sight targets at intervals (of <30 m) along
the channel. Absolute distances and azimuths were validated using a tape measure and protractor, respectively. Ice temperature
of the channel wall was measured at ~50 m intervals along the survey track using a BIOS digital contact thermometer.

Samples were collected from the subglacial pond for water isotope, biogeochemical, and microbial analyses at four locations
(S6-S9; Figure 2). The first location (S6) was at the pond edge and was comprised of wet ice, which was collected into a whirl-
pak® bag using a flame-sterilized ice axe. Though this sample was ice, we consider it to be affiliated with the pond water since
it was situated in a depression that would receive continual seepage of water from other areas of the pond, and where
temperatures would be sufficiently cold to freeze it relatively fast and completely. The other three sample sites were collected
at intervals (~ 25 m apart) into the pond. At each location, pond slush was scooped into a glass beaker that was acid-washed,
rinsed with 18.2 MΩ cm$^{-1}$ deionized water (DIW), and furnaced (450°C for 4 hrs). Pond ice and water were separated by
decanting the two fractions into separate whirl-pak® bags. Water samples were immediately frozen, and all samples were kept
at -20°C until analysis.

Basal ice from the channel walls and ice from frozen sections of the channel floor were collected at five locations at 50-100 m
intervals (S1-S5; Figure 2). Ice along the channel floor was generally clear and had the appearance of frozen meltwater, in
contrast to the banded opaque glacier and debris-rich basal ice exposed in the tunnel walls. We expect that stagnant late-season
runoff occupied the subglacial channel floor at the end of the melt season, and while some of this water remained liquid near
the endpoint (pond), it froze along the outer section of the channel at the end of the melt season, preserving its microbial and
biogeochemical characteristics. Since freezing would have propagated from the surface to the bottom of the water column
introducing a biogeochemical gradient in the ice, bulk samples were collected along the vertical profile of the channel ice at
each site. Channel ice samples were collected into whirl-pak® bags using a flame-sterilized ice axe.





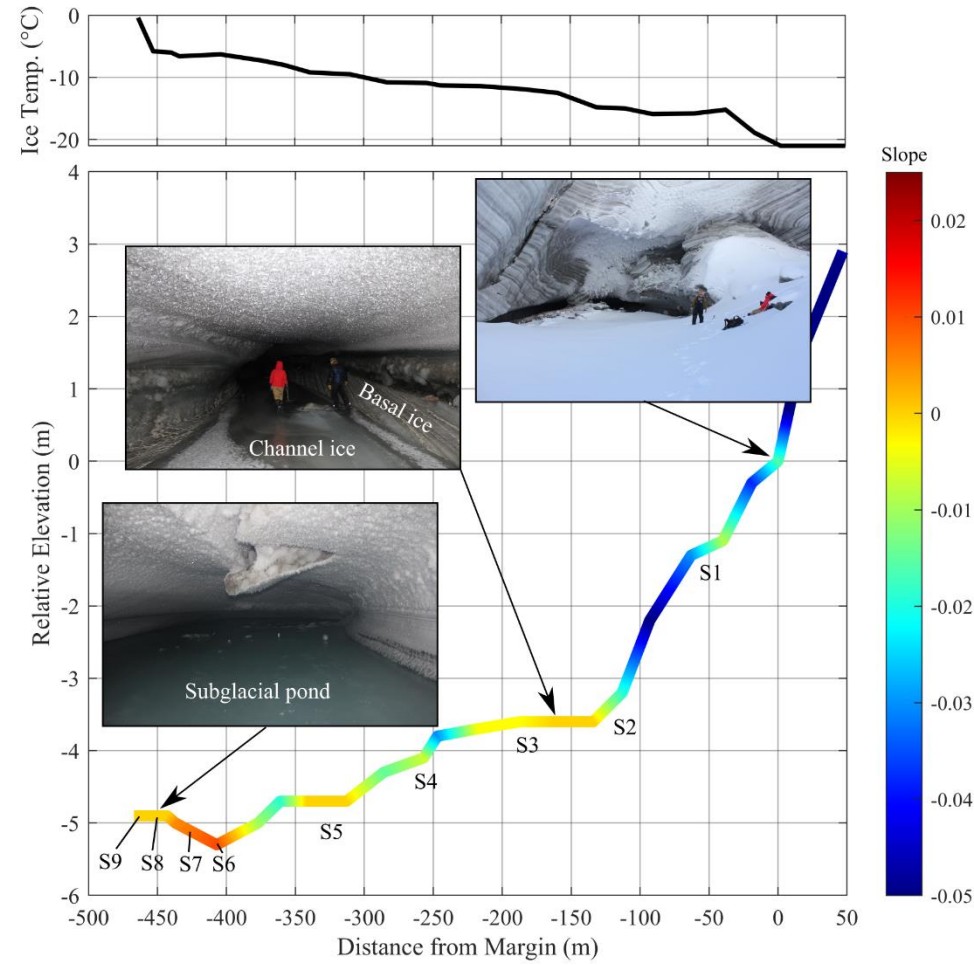

Figure 2: Ice temperature (top) and relative elevation, extent, and slope of the Sverdrup subglacial channel including sample locations of basal ice and channel ice (S1-S5), pond edge ice (S6) and pond slush water and ice (S7-S9) (bottom). Note vertical exaggeration of 50:1.

## 2.3 Laboratory analyses

Samples were melted in a laminar flow hood, equipped with a HEPA filter, at <4°C, in acid-washed (10% HCl for 24 hr), DIW-rinsed and furnaced (450°C for 4 hrs) glass beakers, covered with furnaced aluminum foil. All glassware was acid-washed, DIW-rinsed, and furnaced before use. GF/F filters were also furnaced, and non-sterile plastic-ware was acid-washed, DIW-rinsed, and autoclaved before use. Glassware, filter papers, and plastic-ware were all sample-rinsed before aliquots were collected for analysis, as follows:





- A 10 ml sample was collected in a 25 ml glass beaker for immediate measurement of pH using a Fisherbrand™ accumet™ liquid-filled pH/ATC electrode (13-620-530A). One ml was also collected for immediate measurement of electrical conductivity (Oakton®).

- A borosilicate glass filter tower (Fisherbrand™) and 0.7 μm 25 mm GF/F filter paper were used to filter aliquots of sample into two 15 ml centrifuge tubes (Cole-Palmer) that were frozen immediately for $NH_4^+$ and $PO_4^{3-}$ analysis, one 20 ml amber glass EPA vial (Fisherbrand™), that was kept at 4°C with no headspace, was used for spectrofluorescence analysis, and one 40 ml amber glass EPA vial that was amended with 5 M HCl to pH = 2, and stored at 4°C until dissolved organic carbon (DOC) analysis.

- A 10 ml sterile BD Luer Lok syringe and 0.2 μm PES syringe filter (Whatman™ Puradisc) was used to filter aliquots of sample into a 15 ml centrifuge tube (Cole-Palmer), which was stored at 4°C until major ion analysis, a 2 ml Eppendorf™ tube, which was stored at 4°C until Si analysis, and a 2 ml Fisherbrand™ cryogenic storage vial, which was frozen until water isotope analysis.

- For debris-rich (basal ice) samples, 10 ml of sample was collected in a 15 ml Falcon® tube, 5 ml of a cell-sediment separation detergent (Morono et al., 2013) was added and the mixture was vortexed for 1 minute, then centrifuged
(500 xg) until the sediment separated. The supernatant (for basal ice samples) or 15 ml of water (for all other samples) was transferred to a new 15 ml Falcon® tube, stained with SYBR™ Gold Nucleic Acid Gel Stain (Invitrogen), and then filtered using a borosilicate glass filter tower (Fisherbrand™) and 0.2 μm MF-Millipore™ filter membrane. Filter papers were mounted on a microscope slide and 20 fields of view were counted at 100x magnification using a Nikon Eclipse 80i microscope.

- Remaining sample was filtered through Pall Supor ® 47 mm filter papers using a Thermo Scientific™ Nalgene filtration kit. DNA was extracted from these filter papers using a FastDNA SPIN Kit for Soil (MP Biomedicals, Irvine, CA, USA) by loading the filter paper directly into the lysis tube then following manufacturer's instructions. The DNA extracts were quantified using a Qubit HS DNA kit (Invitrogen, Waltham, MA, USA). The 16S rRNA gene was amplified in triplicate from each sample with 30 cycles of PCR using modified universal primers (515F (Parada et
al., 2016) and 806R (Apprill et al., 2015)). The triplicate reactions were pooled, and Illumina sequencing adapters were added with an additional 8 cycles of PCR, in triplicate. The triplicate adapter reactions were pooled for each sample and purified using a Wizard PCR clean-up system (Promega, Madison, WI, USA) then sequenced at the University of Wisconsin Genomics Core Facility using Illumina MiSeq technology.

## 2.4 Sample measurements

Water isotopes were measured using a Los Gatos Research Liquid Water Isotope Analyzer (LWIA-45-EP), and calibrated with USGS reference standards. Water isotopes are reported by reference to the Vienna standard mean ocean water (VSMOW), in δ notation (Pinti, 2011). Analytical reproducibility for $δ^2H$ and $δ^{18}O$ was 0.6‰ and 0.4‰, respectively.



Samples were analyzed for major ions (F⁻, Cl⁻, Br⁻, NO₃⁻, SO₄²⁻, Na⁺, Mg²⁺, and Ca²⁺) by ion chromatography using a Metrohm Compact IC Flex ion chromatograph equipped with a C4 cation column and an aSupp5 anion column. Determination of
reactive phosphorus and ammonium were measured using principles of spectroscopy following methods outlined by (Strickland and Parsons, 1972) and a Thermo Scientific ᵀᴹ Genesysᵀᴹ 150 UV-Visible Spectrophotometer with a 10 cm path length quartz cuvette. Silica was also determined using this spectrometer, but with 1 cm path length disposable cuvettes, following methods for heteropoly blue (APHA et al., 2017). Precision and accuracy were better than 5% for all analyses. Non-purgeable DOC was measured by high temperature combustion with a Shimadzu TOC-V (CPH) analyzer equipped with a high
sensitivity platinum catalyst. We used fluorescence spectroscopy to characterize fluorescent dissolved organic matter (FDOM) using a HORIBA Aqualog with a quartz cuvette (1 cm path length) and a xenon lamp as an excitation source. Excitation was measured at 5 nm intervals from 230 nm to 600 nm and emission was measured every 2.33 nm from 245 nm to 826 nm, using an integration time of 10 seconds to produce excitation emission matrices (EEMs) for samples and blanks.

**2.5 Data Analysis**

**2.5.1 Parallel factor analysis**

We used parallel factor analysis (PARAFAC) to decompose the EEMs and identify/quantify FDOM characteristics (Murphy et al., 2013). PARAFAC was completed using the N-way and drEEM toolboxes in Matlab (2021), following methods described by Murphy et al (2013). EEMs were corrected for instrument bias, inner filter effects, and regions of scatter were excised after a DIW blank was subtracted from the measured sample. Components derived from PARAFAC modeling were compared to
those identified in other studies using the online spectral library of auto-fluorescence by organic compounds in the environment, OpenFluor (Murphy et al., 2014).

**2.5.2 Bioinformatics**

The 16S rRNA gene amplicon sequences were processed using *mothur* (Kozich et al., 2013) to filter reads that did not meet minimum quality control thresholds (maxambig=0, maxlength =315, maxhomop=8), join paired reads, cluster sequences into
operational taxonomic units (OTUs) using 97% sequence identity, and assign taxonomic identification to OTUs using the SILVA SSU database v138. A total of 2868 singleton OTUs, or those that appeared once across the entire dataset, were removed prior to further analysis. Diversity calculations were completed using the phyloseq package (v 1.28.0) in R (v 3.6.0).

**2.5.3 Water isotope model**

Using the principles of isotopic fractionation, a model was developed to estimate the isotopic composition of incremental ice,
incremental vapor, and residual water as a hydraulically isolated waterbody progressively freezes and evaporates (Supplementary Methods S1). The degree to which isotopes fractionate during freezing depends on the rate of freezing and can be described by Eq. (1):



$$\varepsilon_{i-w} = \delta_i - \delta_w = 1000(\alpha - 1) \qquad (1)$$

Where $\delta_i$ and $\delta_w$ are the isotopic composition of the ice and water, respectively, $\varepsilon_{i-w}$ is the isotopic difference between ice

and water, and $\alpha$ is the respective equilibrium fractionation factor. The degree to which isotopes fractionate during evaporation

($\varepsilon_{w-v}$) depends on the temperature of the water during evaporation and can also be described by Eq. (1) using the isotopic

composition of water and vapor instead of ice and water, respectively.

The isotope model equations are described in detail in Supplementary Methods S1. Fixed variables included: 1) the initial

isotopic composition of the waterbody, which was set to the average isotopic composition of channel ice samples, 2)

equilibrium values for evaporation, which were set to experimental values obtained for evaporation at 0°C (Majoube, 1971),

3) equilibrium values for freezing, which were set to the average of the isotopic fractionation between ice and water in pond

slush, and 4) the step interval, which was set to simulate 0.1% of the waterbody freezing in each step of the model. We adjusted

the rate of evaporation vs freezing, to optimize the model's fit with the observed $\delta^2$H-$\delta^{18}$O of channel ice and pond ice along

the incremental ice line, and the pond water samples along the residual water line (Figure S2). We then fixed the evaporation

vs freezing rate and ran the model to estimate the stage of freezing (i.e. the fraction of residual water vs frozen and evaporated

water) at each sample site in the pond. To do so, the observed $\delta^2$H-$\delta^{18}$O of ice and water samples at each site in the pond were

compared to the modeled incremental ice line and the modeled residual water line, respectively. The step with the most similar

$\delta^2$H-$\delta^{18}$O was identified for each site and the corresponding stage of freezing was determined.

### 2.5.4 Cl⁻ model

Cl⁻ does not readily precipitate as salts, interact with rocks, or become assimilated in significant quantities by microorganisms

(Davis et al., 1998). Since there were no identifiable sources or sinks for Cl⁻ in the system, we used the principles of solute

fractionation to develop a model that estimates the [Cl⁻] in incremental ice and residual water as an isolated waterbody

progressively freezes and evaporates. Cl⁻ is largely rejected from the ice crystal lattice during freezing (Killawee et al., 1998;

Clayton et al., 1990) and from water vapor during evaporation, and thus it concentrates in the residual water. Though effectively

all non-volatile solutes are excluded from water vapor, the degree to which solutes are excluded from the ice is described by

the effective segregation coefficient ($K_{eff}$) in Eq. (2):

$$K_{eff(X)} = \frac{[X]_{ice}}{[X]_{water}} \qquad (2)$$

Where $K_{eff(X)}$ is the effective segregation coefficient for biogeochemical species $X$, and $[X]_{ice}$ and $[X]_{water}$ are the

corresponding concentrations of that species in coincident ice and water samples.

The Cl⁻ model equations are described in detail in Supplementary Methods S2. Fixed variables included: 1) the initial [Cl⁻] of

the waterbody, which was set to the average [Cl⁻] of channel ice (Simulation 1) or with the addition of Cl⁻ affiliated when 5%

(Simulation 2) or 15% (Simulation 3) of the initial solute is comprised of basal solute sources (represented by the average

solute concentrations in basal ice samples), 2) $K_{eff}$, which was derived from coincident ice and water samples in the pond slush





using Eq. (2), 3) the step interval, which was set to simulate 0.1% of the waterbody freezing in each step of the model, and 4)
the rate of evaporation vs freezing, which was set to the ratio derived from isotope modeling.

Results from the Cl⁻ model were used to estimate the stage of freezing (i.e. the fraction of residual water vs frozen and
evaporated water) at each sample site in the pond (Figure S2). To do so, the observed [Cl⁻] in ice and water at each pond sample
site were compared to modeled [Cl⁻] in incremental ice and residual water, respectively. The step with the most similar [Cl⁻]
was identified for each site and the corresponding stage of freezing was determined.

### 235   2.5.5 Geochemical model

Similar to Cl⁻, other dissolved impurities are excluded from water vapor during evaporation and the extent to which they are
also rejected from the ice crystal lattice during freezing can be quantified using Equation 2 (Killawee et al., 1998; Clayton et
al., 1990). We therefore used the same theory and equations as the Cl⁻ model to produce a geochemical model that simulates
the concentration of other solutes and impurities in residual water and incremental ice (Supplementary Methods S2; Figure
S2) at the stage of freezing affiliated with each sample site (determined from the Cl⁻ model). We then compared measured
concentrations at each pond sampling site to modeled concentrations. Since the model only simulates the effects of freezing
and evaporation, we interpreted $[X]_{measured} > [X]_{modeled}$ as evidence for potential net sources of chemical species '$X$' and
$[X]_{measured} < [X]_{modeled}$ as evidence for potential net sinks of chemical species '$X$'.

### 3 Results

### 245   3.1 Physical system

The ponded water from which samples were retrieved was located at the endpoint of a remnant subglacial tunnel that extended
467 m horizontally into the glacier at an angle of ~18° relative to the direction of ice flow (NNW; Figure 1). The location of
the ponded water was therefore ~150 m in perpendicular distance from the margin and ~79 m below the glacier surface (Figure
1). Along its 467 m length, the channel floor dropped 5.3 m in elevation (Figure 2) and terminated at an ice-wall that is located
250   near the edge of a subglacial 'cliff', which drops almost 200 m over a horizontal distance of ~150 m towards the glacier
centreline (Figure 1c). The glacier bed reaches a maximum depth of -190 m a.s.l. (or ~300 m below the glacier surface).

The ambient air temperature at the surface of the glacier remained < 0 °C from mid-September 2018 until fieldwork in May
2019, with temperatures falling to < -30 °C in 6 of these months (Figure S1). At the time of the field survey, ice temperatures
along the channel wall increased from -21 °C at the tunnel entrance to -0.4 °C at the endpoint, indicating the presence of a
255   subglacial heat source. The channel floor was completely frozen from the entrance to ~ 405 m (Figure 2). Liquid water was
observed on the channel floor from ~ 405 m and transitioned into a slushy pond, which extended to the endpoint at 467 m. A
higher water:ice ratio was observed in the pond with distance towards the endpoint. The surface of the slushy pond increased
in elevation by 0.4 m towards the endpoint (Figure 2). Hydraulic head at the ice-bed interface in the vicinity of the pond was
higher than at any other point across the glacier cross-section (Supplementary Methods S3 and Figure S3).





### 3.2 Water isotopes

The isotopic composition of ice samples from the channel floor (S1-S5) lie below the local meteoric water line (LMWL) for the Sverdrup catchment (Copland et al., 2021), which is consistent with the isotopic composition of meltwater originating from local snow/glacier ice. Channel pond and basal ice samples fall above the LMWL and exhibit more negative $\delta^{18}O$ and $\delta^{2}H$ values relative to the channel ice. The difference between respective ice and water samples at sites 8 and 9 was 1.8-1.9 ‰ for $\delta^{18}O$ and 10.9 ‰ for $\delta^{2}H$, which is consistent with equilibrium values measured in other slow-freezing environments (Table S1) (Jouzel et al., 1999; Ferrick et al., 2002).

Our isotope modelling experiments indicate that a freeze:evaporation rate of ~40:1 was required to best fit the model to the observed water and ice data points (Figure 3a). This model also indicates that ~10% of the initial water remained residual at pond sampling sites 8 and 9 (Figure 3c). The $\delta^{2}H$ and $\delta^{18}O$ values at Site 7 show that a similar fraction of water remained residual. The $\delta^{2}H$ and $\delta^{18}O$ values at Site 6 showed the most extensive freezing, with only 2% remaining as residual water (Figure 3c).

### 3.3 Cl⁻ concentrations

An average $K_{eff(Cl^-)}$ value (Equation 2) of 0.22 was derived from coincident pond ice (2.0 μM and 2.2 μM) and water (15.8 μM and 15.7 μM) (Figure 4) samples at sites S8 and S9, respectively. This $K_{eff(Cl^-)}$ falls within the range observed in other studies (Santibáñez et al., 2019). The average [Cl⁻] in channel ice samples was 2.1 μM, which is consistent with concentrations in regional glacier ice and snow (Dubnick et al., 2020). Cl⁻ fractionation modeling (Supplementary Methods S1) of water with initial [Cl⁻] of 2.1 μM suggests that ~9% of the water in the pond at S8 and S9 remained as residual to yield the [Cl⁻] measured at these sites. These results are in close agreement (within 1%) with corresponding results from the isotope fractionation model at S8 and S9 (Figure 3c). Cl⁻ comprised a very small portion of the solutes in basal ice, so simulations that included contributions of basal solutes (up to 15% of the solute load) had little effect (<0.1%) on the estimated stage of freezing at each site.





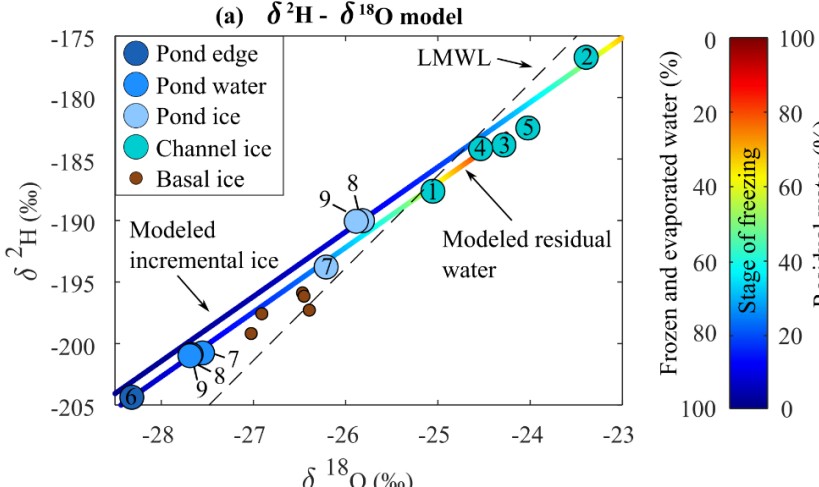

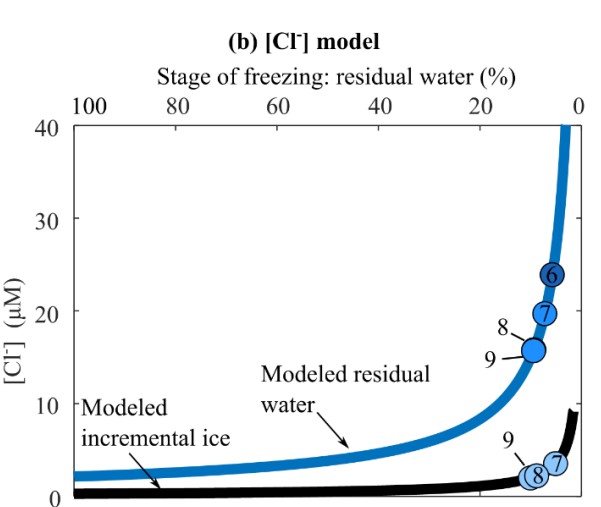

Figure 3: (a) δ2H- δ18O plot showing the modelled residual water and incremental ice as water progressively freezes and evaporates, as well as measured concentrations for channel ice and pond samples. LMWL is from Copeland et al. (2021) (b) modelled Cl- concentrations in residual water and incremental ice as water progressively freezes and evaporates, as well as measured concentrations for channel ice and pond samples fit to respective lines, and (c) the average corresponding percent frozen and evaporated for ice and water samples at sampling sites S6, S7, S8 and S9, derived from the isotope model in (a) and the [Cl-] model in (b). Error bars indicate the minimum and maximum modelled results for the ice and water samples at each location.

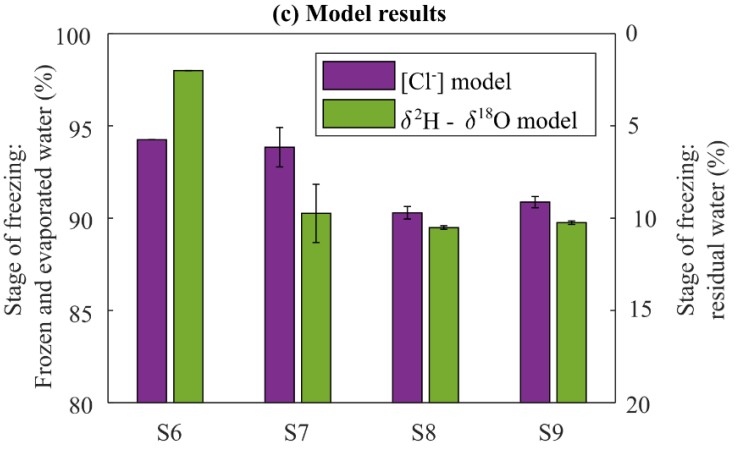



Figure 4: Biogeochemical content of basal ice, channel ice and pond water/ice.





### 3.4 Major ion chemistry

Channel ice samples from S1-S5 were generally dilute ($\bar{x}$ = 0.91 mg L$^{-1}$) but displayed considerable variability in the concentration of total measured solutes. Pond ice samples (at sites 8-9) had similar total measured solute concentrations (1.1 and 1.2 mg L$^{-1}$) but coincident pond water samples had concentrations ~7-fold higher (7.0 mg L$^{-1}$ and 7.2 mg L$^{-1}$, respectively). Pond edge ice (site 6) had the highest total measured solute load of all channel samples (11.6 mg L$^{-1}$). Basal ice had highly variable total measured solute concentrations (1.5-5.4 mg L$^{-1}$) (Figure 4). $K_{eff}$ values derived using coincident ice and water samples at sites S8 and S9 are <0.3 for each major ion, and consistent with those reported by Santibanez et al. (2019) (Figure S4).

Geochemical modeling (Supplementary Methods S2) using late season runoff as the sole solute source (Simulation 1) underestimated the concentration of many of the dominant solutes in pond water, including $HCO_3^-$, $K^+$, $Mg^{2+}$, $Ca^{2+}$ and $SiO_2$ (Figure 5a). The solute:Cl$^-$ ratios for all biogeochemical species were greater in basal ice than in channel ice (Figure 5b), so simulations that also incorporated basal solute contributions to the pond (comprising 5-15% of the initial solute load; Simulations 2 and 3) generally yielded more accurate concentrations of these solutes (Figure 5b).

### 3.5 FDOM components

PARAFAC modeling resolved two fluorescent components that comprised 92 % of the variability in the dataset (Figure S5). Component 1 (C1) overlaps with ''humic-like'' fluorescence (Coble, 1996) and has been associated with fulvic acid DOM fractions (Ohno and Bro, 2006). While this fluorescence feature is not consistently identified in glacier ice, similar fluorescence signatures are ubiquitous in a wide range of terrestrial (Ohno and Bro, 2006; Yamashita et al., 2011) and marine (Jørgensen et al., 2011; Retelletti Brogi et al., 2018; Walker et al., 2009) environments. Further, this fluorescence component has been identified in Antarctic watersheds without higher plants (Cory and McKnight, 2005; Barker et al., 2013) and has been correlated with microbial activity in ocean waters (Jørgensen et al., 2011) indicating that it may also be produced by microbial reworking of organic matter (OM). Further, this fluorophore is persistent in many environments suggesting that, once formed, it may not be readily altered (Yamashita et al., 2010).

Component 2 (C2) overlaps with the fluorescence of amino acids, in particular as either free or bound tyrosine (Peak B, Coble, 1996). Similar FDOM is widely detected in glacial ice, including englacial and basal ice from Greenland, Antarctica, and across the Canadian Arctic from the supraglacial snowpack to ice dating back to the Last Glacial Maximum (D'Andrilli and McConnell, 2021; D'Andrilli et al., 2017; Pautler et al., 2012; Dubnick et al., 2010, 2020; Barker et al., 2006, 2009; Fellman et al., 2010). Tyrosine-like fluorescence is commonly associated with OM of low molecular weight and aromaticity and chemical species that are readily degraded by microorganisms (Coble, 2014). Unlike C1, C2 is rapidly transformed in proglacial and downstream aquatic systems (Wu et al., 2003; Saadi et al., 2006; Fellman et al., 2008; Barker et al., 2013).

1⊥

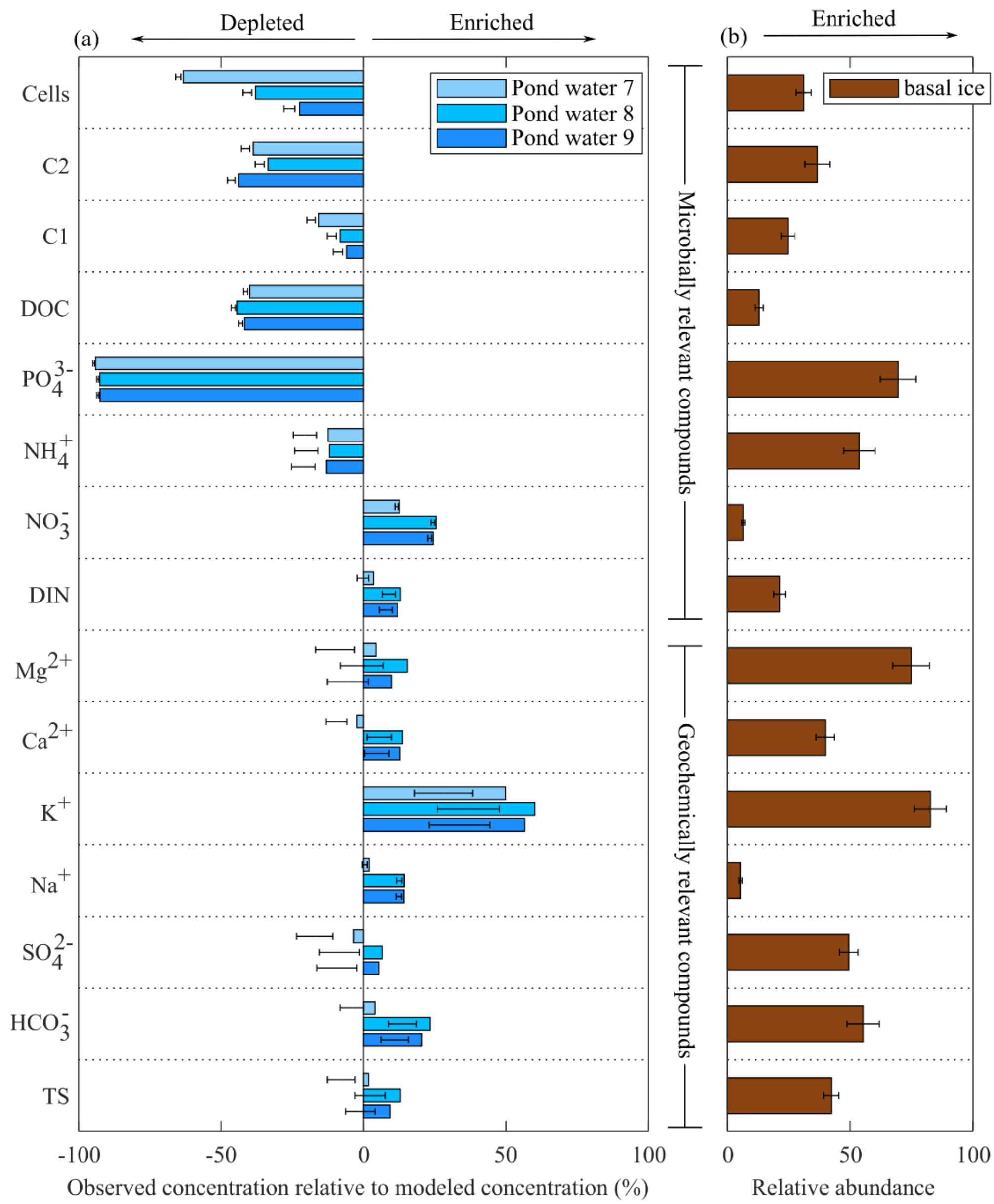


Figure 5: (a) geochemical modelling results compared to observed concentrations in pond samples. Bars represent simulations that assume channel water as the sole water and solute source (Simulation 1) while error bars represent simulations with an additional 5-15% solutes from basal ice (Simulations 2 and 3). (b) provides a measure of the relative abundance of the different solutes in basal ice compared to channel ice according to the formula y = $\frac{X_{BI}:Cl^-_{BI}}{X_{CI}:Cl^-_{CI}}$ where X is the average concentration of an
analyte in either basal ice (BI) or channel ice (CI) and $Cl^-$ is the average chloride concentration, and y is the relative abundance of that analyte. Error bars represent 0.1 standard deviations.



### 3.6 Nutrients

Channel ice samples contained relatively low concentrations of organic and inorganic nutrients, including $NO_3^-$ ($\bar{x}$ = 0.68 μM),
$NH_4^+$ ($\bar{x}$ = 0.31 μM), $PO_4^{3-}$ ($\bar{x}$ = 0.03 μM), and DOC ($\bar{x}$ = 5.8 μM) (Figure 4). Concentrations of these nutrients were all higher
in the pond water than in coincident pond ice samples, yielding $K_{eff}$ values < 1 (Figure S4). Compared to modeled
concentrations, pond waters were depleted in all nutrients (i.e., $NH_4^+$, $PO_4^{3-}$, DOC, and DOM components C1 and C2) except
$NO_3^-$ (Figure 5). Basal ice had higher nutrient:$Cl^-$ ratios than did the channel ice, especially in the case of $PO_4^{3-}$ and $NH_4^+$
(Figure 5b). Consequently, simulations that involve basal solute contributions to the pond (Simulations 2 and 3; Figure 5b)
increase the amount by which these nutrients appear depleted in pond water (Figure 5a).

### 3.7 Microbiology

Cell concentrations in channel ice averaged 5.2 x $10^4$ cells $mL^{-1}$ while concentrations in the pond water were 6.3 – 11.9 x $10^4$
cells $mL^{-1}$, comparable to the range found in other subglacial waters (Christner et al., 2014; Gaidos et al., 2004; Mikucki et al.,
2009). Over 85% of the 16S rRNA gene sequences in pond water samples were affiliated with one of six OTUs that fell in the
class Gammaproteobacteria: *Massilia* (28-40% of total reads), two *Polaromonas* OTUs (25-27%), an unclassified
Oxalobacteraceae (14-18%), *Rhoderax* (5-15%), and *Undibacterium* (3%) (Figure 6). While these OTUs (except the
unclassified Oxalobacteraceae) were all detected in basal ice, they were more abundant in channel ice samples (including the
unclassified Oxalobacteraceae), comprising 23-32% of the assemblage (Figure 6). Their higher relative abundance in channel
ice suggests these organisms probably originated from late season runoff, rather than from subglacial sources. Chao1,
Shannon's (H) and Simpsons (D) diversity indices all indicate taxonomic diversity was lower in pond water than in channel
ice or basal ice (Figure 6).





Figure 6: (a) Chao1, (b) Shannon's and (c) Simpson's diversity index for microbial assemblages in basal ice, channel ice and pond water samples. (d) The composition of microbial assemblages detected in each sample. Assemblages are grouped by phylum, except for the 6 most dominant OTUs in the pond water, which were all Gammaproteobacteria and are further classified here by genus (i.e. *Massillia*, *Polaromonas* (x2), an unclassified Oxalobacteraceae, *Rhodoferax*, and *Undibacterium*). Insufficient sample was available for Channel Ice S2, Pond Edge S6, and Pond Water S7.



## 4 Discussion

### 4.1 Evolution of the subglacial tunnel and pond

Subglacial channels develop when inflowing waters have sufficient heat and pressure to melt ice at a rate that exceeds that of creep closure from the overlying ice (Röthlisberger, 1972; Nye, 1976). These channels grow during spring-summer to deliver meltwater to the glacier terminus and are closed by ice creep as runoff rates recede in the fall. Closure rates are typically lower near the ice margin where overlying ice is thin and overburden pressure is low and are faster at internal locations where overlying ice is thick and overburden pressure is high (Röthlisberger, 1972). At the time of the field survey, the subglacial

tunnel terminated at approximately the upper edge of a 'cliff', or rapid deepening along the glacier bed profile (Figure 1). Rates of creep closure would be much higher at internal locations beyond this subglacial 'cliff' due to the ice overburden pressure (Figure S3). Creep closure near the pond was likely responsible for squeezing the subglacial pond slush into a mound, resulting in the positive slope of the pond surface in this area at the end of winter (Figure 2).

As ambient air temperatures fell towards the end of the melt season of 2018 (remaining below 0°C from September 2018 –

June 2019; Figure S1) drainage into the subglacial channel from the ice margin would have ceased, leaving ponded water along the channel floor in areas with low slope. The cold, dense ambient air drained into the tunnel, progressively freezing the stagnant water on the channel floor. Although the freezing process along the channel floor may have released some latent heat, additional heat source(s) would have been required to maintain liquid water and the near 0°C ice and air temperatures that were observed at the tunnel endpoint in May. The Sverdrup Glacier is marine-terminating ~3 km downstream of the subglacial

tunnel and displays a retrograde subglacial slope (Paden et al., 2019), resulting in ice that is grounded below sea level across most of the glacier's width near the subglacial tunnel (Figure 1). This geometry allows heat from the ocean to more effectively warm areas of the glacier bed. Additional heat may also originate from friction near the glacier bed. Prior studies suggest regions of the Sverdrup Glacier have flow regimes that involve enhanced deformation of basal ice (Van Wychen et al., 2017), which produce frictional heat. Geothermal heat may also contribute to the subglacial heat flux but the regional geothermal heat

flux is relatively low (65 +/- 5 m W m$^{-2}$; (Grasby et al., 2012)).

### 4.2 Concentration effects

The water isotope and Cl$^-$ models produced consistent estimates of the extent of freezing at the pond sampling sites (Figure 3c). The portion of residual water increased with distance into the pond, from 6% remaining as residual water at the pond edge (S6) to 9% remaining as residual water at S9. Sites S6 and S7 are at lower elevations than S8 and S9 (Figure 2), and would

have experienced colder overlying air in this depression. Freezing processes (and to a lesser degree, evaporation) were, therefore, the dominant controls on the concentration of solutes in the pond, resulting in concentrations up to 7 times those observed in late-season runoff.

The temperature gradient along the downward-sloping subglacial tunnel (Figure 2) facilitates the drainage of cold, dense ambient air into the tunnel and drainage of warm buoyant air out of the tunnel. Regional air masses are cold and have low





relative (65%) and specific (0.2 g kg$^{-1}$) humidity during the winter (Vincent et al., 2007). The convection and warming of this ambient air through the tunnel would therefore facilitate continual evaporation from the subglacial pond throughout the cold season. The tunnel showed evidence of evaporation; the roof/upper section of the walls were covered in a blanket of intricate ice crystals formed by condensation of water vapor as the buoyant/warm/moist air mass traveled out of the tunnel and cooled. During evaporation, solutes are preferentially excluded from the vapor phase, so evaporation at the pond surface increased

solute concentrations in the residual water.

The drainage of cold, dense air along the tunnel floor would also have frozen the late-season runoff pooled on the channel floor, from the surface to the bed, and from the channel entrance towards the endpoint. The pond was comprised of a slushy mixture of ice and water, indicating a slow freezing rate (Michel and Ramseier, 1971) near the tunnel endpoint, and isotope modeling suggests that freezing occurred at a rate ~40 times that of evaporation. Suspended and dissolved impurities and light

water isotopes are preferentially rejected from ice crystals during the freezing process, resulting in residual water that becomes increasingly concentrated in impurities (Clayton et al., 1990; Killawee et al., 1998) and isotopically light.

### 4.3 In-situ geochemical sources

Unlike the waters contained beneath many other polythermal glaciers, our geochemical modeling simulations suggest that the pond only received small (~5-15%) contributions of solutes from basal sources. As discussed above, the channel floor ice and

pond water originated as late-season runoff that pooled along the channel floor in areas of low slope. Compared to geochemical modeling that simulates late-season runoff as the sole solute and water source to the pond, pond water samples were enriched in total solutes, including $SO_4^{2-}$, $K^+$, $Ca^{2+}$, $Mg^{2+}$ (Figure 5a) and Si. In glacial environments, these elements are commonly derived from rock-water reactions (Tranter et al., 2002) and are found at high concentrations in basal ice (Dubnick et al., 2020) and subglacial discharge (Wadham et al., 2010; Li et al., 2022). Importantly, these solutes were enriched in basal ice samples

in this study (Figure 5b), suggesting they are liberated from basal processes.

Basal solute contributions to the pond most likely originated as seepage from basal ice along the pond perimeter. The interstitial water content of basal ice scales with temperature, so the relatively warm ice in this area (Figure 2) could promote drainage of solute-rich water into the pond. Though the pond could also receive basal solutes from a distributed drainage system at the ice-bed interface, hydraulic modeling suggests that if waters occupied a distributed drainage system, they were more likely to

drain away from the pond than towards the pond (Figure S3). The pond could also acquire basal solutes by rock-water interactions at its base. However, this channel receives high volumes of seasonal meltwater over consecutive years, so reactive components of the bedrock would be depleted (including those with slow reaction kinetics).

Simulations that incorporated basal solutes improved the accuracy of modeled solute concentrations, though the accuracy of the models were inconsistent among geochemically-relevant compounds (Figure 5a). The solute composition of basal ice,

subglacial water, and species liberated from *in situ* rock weathering, can vary dramatically within a subglacial catchment (Dubnick et al., 2020; Yde et al., 2010; Tranter et al., 2002) due to variation in the composition of the underlying bedrock, redox conditions, ice/water temperatures, and freeze-thaw histories. The five basal ice samples explored in this study also



showed high variability in solute chemistry (Figure 4), confirming that potential basal solute sources in the subglacial tunnel are heterogeneous and that our model may not accurately represent the precise composition of the basal solutes entering the pond.

## 4.4 Microbiology and biogeochemical nutrient sinks

Sequencing of 16S rRNA gene amplicons indicates that the microbial assemblage in the subglacial pond was dominated by organisms in the order Burkholderiales, which are commonly found in subglacial systems (e.g. Cheng and Foght, 2007; Achberger *et al.*, 2016; Dubnick *et al.*, 2020). More specifically, the most dominant OTUs, *Massilia*, *Polaromonas*, *Rhodoferax* and *Undibacterium* (Figure 6) are all related to microbes detected at high abundance in other glacial environments (Foght et al., 2004; Lanoil et al., 2009; Darcy et al., 2011; Mitchell et al., 2013; Perini et al., 2019; Dubnick et al., 2020; Dunham et al., 2021) including supraglacial, proglacial, and/or subglacial aquatic systems worldwide. Moreover, these OTUs are related to psychrophilic or psychrotolerant species (e.g. Foght *et al.*, 2004; Darcy *et al.*, 2011; Wang *et al.*, 2018; Perini *et al.*, 2019), and many of these taxa have been shown to have a range of structural and functional adaptations for survival in cold temperatures (Margesin and Miteva, 2011).

All six dominant OTUs in the pond were closely related (>98%) to globally distributed environmental sequences and cultured representatives, suggesting these organisms are habitat generalists with broad environmental tolerances. For example, *Polaromonas* are dominant in polar and high-elevation environments (Darcy et al., 2011), are thought to rapidly evolve to new environments through extensive horizontal gene transfer (Yagi et al., 2009), and are considered metabolically diverse "opportunitrophs" (Meyer et al., 2004; Polz et al., 2006). The meltwaters that drained into the subglacial system towards the end of the melt season originated from cryoconite holes, supraglacial streams, englacial ice, precipitation, and extraglacial aquatic or terrestrial sources. These environments are exposed to distinct nutrient pools, redox conditions, and, in some cases, high levels of solar radiation and/or warmer temperatures. The shift in environmental conditions to those in the cold, dark, oligotrophic subglacial pond may have decimated populations not capable of survival in this very different system, selecting for generalist organisms that are better adapted to those conditions (Xu et al., 2021).

Consistent with the notion of being habitat generalists, the six dominant OTUs in the pond water are closely related to genera with extreme metabolic versatility. For example, despite sharing nearly identical 16S rRNA genes, *Polaromonas* strains from different glaciers can have very different phenotypes (Gawor et al., 2016), with some strains variably showing evidence of an ability to oxidize $H_2$ (Sizova and Panikov, 2007; Dunham et al., 2021), arsenite (Osborne et al., 2010), and/or organic matter (Jeon et al., 2003; Mattes et al., 2008; Osborne et al., 2010). Furthermore, several cold-tolerant species of *Rhodoferax* (e.g. *R. antarcticus* and *R. fermentans*) have been shown to be facultative anoxygenic phototrophs that can grow photoheterotrophically (using a variety of organic and fatty acids or glucose) and photoautotrophically (using $CO_2$) or that can grow via aerobic respiration in dark environments (Madigan et al., 2000). *R. fermentans* has also been shown to be capable of nitrate reduction (Hougardy and Klemme, 1995) and a close relative of *R. ferrireducens* from glacial sediments in Iceland has been shown to fix $CO_2$ with energy derived from the $H_2$-ferric iron redox couple (Dunham et al., 2021). The high abundance of *Rhodoferax*



in the pond may be due to its metabolic flexibility which allows it to compete for nutrients despite their changing availabilities as they are transported from the supraglacial/extraglacial to the subglacial system, and in the subglacial pond as it becomes increasingly cold, concentrated in solutes, and depleted in labile nutrients over the winter.

Our microbial and biogeochemical datasets both suggest the pond functioned as a hotspot for C-cycling over winter. We measured ~43% less DOC than our modeling predicted and a more depleted reservoir of the labile tyrosine-like (C2) fluorophore, which is often rapidly transformed in proglacial and downstream aquatic systems (Wu et al., 2003; Saadi et al., 2006; Barker et al., 2013; Fellman et al., 2010) (Figure 5). Previous studies of *Polaromonas*, *Rhodoferax*, and *Massilia* strains, which comprised ~70% of the pond microbial assemblage, suggest they are fuel their metabolisms by coupling the oxidation of organic matter with reduction of $O_2$ (Hougardy and Klemme, 1995; Jeon et al., 2003; Mattes et al., 2008; Osborne et al.,

2010), and thus could be responsible for depletion of DOM. Assuming 70% of the cells in the pond ($\bar{x} = 7.5$ x $10^4$ cells mL$^{-1}$) were active heterotrophs, and that the size of this population was relatively stable throughout the ~240 days over winter, the observed DOC deficit (3.9 μM) equates to an average rate of C-remineralization of 0.2 fmol C cell$^{-1}$ day$^{-1}$. This rate is comparable to metabolic rates sufficient for microbial growth at ~0°C and is orders of magnitude higher than metabolic rates required for cellular maintenance or survival at 0°C (Price and Sowers, 2004). Though this is only an estimate for C-

remineralization in the pond, rates could be higher if less than 70% of the microbial assemblage was engaged in heterotrophic energy metabolisms, or if additional OC pools were also degraded, such as particulate OM, basally-derived OM, or autochthonous OM (our calculation above only accounts for DOC supplied in the late-season runoff).

Microbial activity in the subglacial pond could also explain the depleted reservoir of other inorganic nutrients, including $PO_4^{3-}$ and $NH_4^+$ (Figure 5). Past work has shown that heterotrophic microbial biomass is C-poor yet P- and N-rich (Makino et al.,

2003; Godwin and Cotner, 2015) relative to many terrestrial DOM sources, and that microbial heterotrophic activity has been linked to a simultaneous assimilation of mineral nutrients (Fenchel and Blackburn, 1979; Martinussen and Thingstad, 1987). While $PO_4^{3-}$ is considered to be a preferred and universal source of phosphorus to microbes (Björkman and Karl, 1994), N can be assimilated as $NO_3^-/NO_2^-$ or as the preferred reduced state, $NH_4^+/NH_3$ (Paul and Clark, 1996; Nyyssönen et al., 2014). Lithoautotrophic activity can contribute to microbial assimilation of inorganic nutrients, including $NH_4^+$ and $PO_4^{3-}$, as has been

demonstrated from laboratory incubations of subglacial sediments and water (Stibal et al., 2012b; Montross et al., 2014; Boyd et al., 2011).

## 5 Conclusions

This research documents the evolution of ponded meltwater and its resident microbial community at the endpoint of a 467 m long remnant subglacial channel through a Canadian Arctic winter. Solute concentrations in the pond were controlled by: 1)

freezing processes, which functioned to cryo-concentrate solutes in the residual water by up to 7 times; 2) seepage of small amounts of basal solutes (comprising <15% of the total solutes) into the pond; and 3) microbial activity, which functioned to deplete the pond's reservoir of the most labile and biogeochemically-relevant compounds, including $NH_4^+$, $PO_4^{3-}$, and DOM.

Sequencing of the 16S rRNA gene revealed decreasing taxonomic diversity among microbial communities with distance into the channel. Six OTUs dominated the microbial community in the pond. These microorganisms likely originated from the extraglacial or supraglacial (rather than subglacial) environment and were related to taxa that are psychrophilic/psychrotolerant, exhibit extreme metabolic diversity, and have broad habitat ranges. Collectively, our findings suggest that generalist microorganisms from the extraglacial or supraglacial environments can become established in subglacial aquatic systems and deplete reservoirs of nutrients over a period of months. The inferred generalist lifestyle of these microorganisms may help them survive the extreme selection pressures imposed by the environment, allowing for not only their persistence but activity. These findings extend our understanding of the microbiology and biogeochemistry of subglacial ecosystems.

**Data Availability**

Sequence data has been deposited in NCBI SRA under BioProject ID PRJNA907039 and geochemical data are available via Zenodo DOI 10.5281/zenodo.7384156.

**Author contribution**

AJD designed and carried out the fieldwork, laboratory work, data analysis, developed the model codes and performed the simulations. RLS extracted DNA from field samples and conducted the bioinformatics, BDD prepared temperature data record, DB prepared radar and laser altimeter transect data, and CD conducted laboratory DOC analysis. AJD, RLS and BDD prepared the manuscript with contribution from all co-authors. MJS, MLS, ESB funded and supervised the work.

**Competing Interests**

The authors declare that they have no conflict of interest.

**Acknowledgements**

We thank the Nunavut Research Institute and the communities of Grise Fjord and Resolute Bay for permission to conduct research on Devon Island, Patrick Williams and Claire Bernard-Grand'Mason for field assistance, the staff at the Polar Continental Shelf Program for field logistics support, Maya Bhatia for laboratory facilities for DOC analysis at the University of Alberta Molecular Biogeochemistry Lab, Montana State University Environmental Analytical Lab for water isotope measurements, and the University of Wisconsin Genomics Core Facility for 16S rRNA gene sequencing.





This research was supported by the Natural Sciences and Engineering Research Council of Canada (Discovery grant no. 05234-2015), the Polar Continental Shelf Program (grant no. 68519), and the National Aeronautics and Space Administration (NNX16AJ64G)

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
