# Peer review of "Biogeochemical evolution of ponded meltwater in a High Arctic subglacial tunnel"

_The Cryosphere, 2022_

## Referee Comment (RC2)

**General comments**

It was a joy to read the manuscript by Dubnick and colleagues. The paper describes in detail the geochemical transformation (evolution) of late-season glacial meltwater as it undergoes freezing subglacially over winter. To my knowledge this is the first study of the type that can provide empirical data on these processes in a very interesting field site that can act as a natural laboratory. Despite the particular nature of the study site, the results presented here are of broader relevance and can inform on a wider array of subglacial systems. They exemplify what microbial and biogeochemical processes can occur subglacially in the context of subglacial environments as being active microbial habitats. They do so in a quantitative way by combining geochemical models, geochemical and microbial observations and go beyond simple data reporting often the case in similar studies. The manuscript is extremely well written, with great context and explanation for both expert and non-expert readers, well within the relevant scopes of the Cryosphere journal.

However, before recommending the article for publication, there are some minor comments that I believe would need addressing or clarifying:

**Specific comments**

1. Fig. 4: Is it possible to add the uncertainty on measurements here? (e.g. Root sum square of precision and accuracy for geochem measurements and 1 or 2 standard dev on cell counts for each sample).

2. I would also add the uncertainty when reporting cell numbers in the text (e.g. section 3.7). I suspect uncertainty in cell numbers might have a greater impact on the enrichment models than for geochem species.

3. Ln 378: Why higher relabund of Gammaproteo in channel ice than rules out other subglacial sources for late-season runoff? The results presented here suggest that incremental freezing might enrich for Gammaproteo. Couldn't the same apply for distributed system subglacial waters (i.e. subglacial sources that might slightly differ from basal ice composition) that would also contribute to late-season runoff? Or are the authors suggesting that these organisms in late-season runoff originate from the supraglacial environment?

4. Ln 433: Can you cite other studies that describe % contribution of basal sources to late-season subglacial residual waters? I would assume late-season runoff to be normally relatively depleted in basal elements (e.g. glacial flour/distributed system input) compared to earlier/peak melt. Not entirely convinced that the pond waters described here are "unlike the waters contained beneath many other polythermal glaciers"; at least late-season residual waters in subglacial channels that is. Or do the authors mean "basal waters from distributed drainage systems"?

5. Ln 434: Perhaps I missed the obvious results (very possible) but why exactly do the geochemical models suggest a maximum basal ice contribution to the pond of 15%? Wouldn't increasing the % contribution of basal ice to the pond better decrease the

enrichment/depletion discrepancy of the model on Fig. 4a? At least for geochemically-relevant species.

6. Ln470: Are you excluding basal waters/melt from other potential sources to late-season runoff? I again think several microorganisms detected in the pond waters might have originated and selected from subglacial sources too (as mentioned on the the first sentence of 4.4 Ln457 and the follow-up paragrap Ln 476-488). I suggest adding "subglacial waters" too to the sentence as potential source. But I agree with the overall interpretation of niche selection.

7. The above comment also makes me caution the conclusion statement that the detected bacterial populations most likely are generalists from supraglacial environments. To me they seem like typical subglacial populations; perhaps generalists yes and ultimately originated from extraglacial systems prior to glacial inception but I'm not sure this is what the authors are claiming (I still think the results are pretty cool though!).

8. There seems to be no discussion on why the cell abundance in the ponded water appears to be depleted relative to model. This seems a bit strange to me. Again, perhaps a higher contribution of basal ice to the model might help explain this (can see technical comments below)?

Data availability:
1. I believe a bit more information is needed regarding the geochemical models used (even if relatively simple). I.e. where/how were they run? Custom scripts? What program (Matlab, R, Exce?). If code was used, it should ideally be uploaded in a repository or included as supp material.
2. Similar comment regarding the bioinformatics: e.g. might it be possible to upload the mothur logfile or command-line summary (e.g. batch file) to a repository or as part of Supp Mat.
3. I'm not sure the Zeonodo link for geochem data is correct and could not access the data (maybe my bad though).

**Technical comments**

Ln 55: should also include Gill-Olivas 2023: https://bg.copernicus.org/articles/20/929/2023/

Methods: consider changing the subscript acronyms in equations to capital letter to avoid confusion with lowercase roman numerals used for "steps" (e.g. $X_{ii(i)}$ for "incremental ice" could become $X_{II(i)}$ etc)

Methods: consider specifying that the geochem model was also applied to cell concentrations

legend of Fig. 3a: isn't inversed? Here shows that red = 100% water and dark-blue = 100% frozen seems opposite to data and theoretical values? (also consider changing rainbow scale to alternative more colour-blind friendly palette (e.g. Viridis))

Fig. 5b: Should x-axis be "relative abundance to channel ice"?

Ln 529: not sure the Zenodo link is correct?

Fig. S3a : I think bed and ice-surface are swapped in the legend. Also what are the capital "A" and "B" above both axes?

---

## Author Comment (AC2)

**General comments**

It was a joy to read the manuscript by Dubnick and colleagues. The paper describes in detail the geochemical transformation (evolution) of late-season glacial meltwater as it undergoes freezing subglacially over winter. To my knowledge this is the first study of the type that can provide empirical data on these processes in a very interesting field site that can act as a natural laboratory. Despite the particular nature of the study site, the results presented here are of broader relevance and can inform on a wider array of subglacial systems. They exemplify what microbial and biogeochemical processes can occur subglacially in the context of subglacial environments as being active microbial habitats. They do so in a quantitative way by combining geochemical models, geochemical and microbial observations and go beyond simple data reporting often the case in similar studies. The manuscript is extremely well written, with great context and explanation for both expert and non-expert readers, well within the relevant scopes of the Cryosphere journal.

However, before recommending the article for publication, there are some minor comments that I believe would need addressing or clarifying:

**Specific comments**

1. Fig. 4: Is it possible to add the uncertainty on measurements here? (e.g. Root sum square of precision and accuracy for geochem measurements and 1 or 2 standard dev on cell counts for each sample).
   Revised as suggested

2. I would also add the uncertainty when reporting cell numbers in the text (e.g. section 3.7). I suspect uncertainty in cell numbers might have a greater impact on the enrichment models than for geochem species.
   Revised as suggested

3. Ln 378: Why higher relabund of Gammaproteo in channel ice than rules out other subglacial sources for late-season runoff? The results presented here suggest that incremental freezing might enrich for Gammaproteo. Couldn't the same apply for distributed system subglacial waters (i.e. subglacial sources that might slightly differ from basal ice composition) that would also contribute to late-season runoff? Or are the authors suggesting that these organisms in late-season runoff originate from the supraglacial environment?

   It is true that subglacial sources could have contributed Gammaproteobacteria, especially if subglacial water had higher relative abundance than the basal ice samples that we used to represent subglacial sources. However, given that we directly observed channel ice to have high relative abundance of Gammaproteobacteria, the simplest explanation is that these microbes originated from late season runoff. Elsewhere in the document (section 4.1 and 4.3), we argue that the geochemistry of channel ice and the physical configuration of the system suggest that subglacial water/basal melt probably had minimal influence on the late season runoff that entered the tunnel. We have clarified this in response to comment 6 below. Regardless, our

interpretation of the origin of these Gammaproeteobacteria is more appropriate for the discussion (section 4.4) than the results, so we have removed this sentence from line 378.

4. Ln 433: Can you cite other studies that describe % contribution of basal sources to late-season subglacial residual waters? I would assume late-season runoff to be normally relatively depleted in basal elements (e.g. glacial flour/distributed system input) compared to earlier/peak melt. Not entirely convinced that the pond waters described here are "unlike the waters contained beneath many other polythermal glaciers"; at least late-season residual waters in subglacial channels that is. Or do the authors mean "basal waters from distributed drainage systems"?

   The contribution of basal sources to late-season runoff would be site specific, since it depends on the configuration of the drainage network, subglacial hydrology, and ice dynamics. It would therefore be extremely difficult to draw comparisons to other studies without extensive hydrological and glaciological context. We are not aware of any other studies that have sampled late-season water that resided in a remnant subglacial channel, let alone studies that are able to constrain the water history.

5. Ln 434: Perhaps I missed the obvious results (very possible) but why exactly do the geochemical models suggest a maximum basal ice contribution to the pond of 15%? Wouldn't increasing the % contribution of basal ice to the pond better decrease the enrichment/depletion discrepancy of the model on Fig. 4a? At least for geochemically-relevant species.

   The geochemical model that assumes late season runoff as the sole solute source of water underestimated the concentration of many dominant and conservative ions in the pond (i.e. the pond was enriched in geochemically-relevant species in Figure 5a bars). Basal ice had high concentration of these solutes, so the addition of up to 15% basal solutes improved modeled concentration of pond water by moving the difference between observed and modeled concentrations towards 0 (i.e. the 'error bars' on figure 5a). The exact proportion of basal ice required to yield model results that match observed concentrations depends on the species and sample, since other minor biogeochemical processes, which are not included in this model, may influence solute concentrations (e.g. contribution of diverse basal sources at different solute ratios, biogeochemical processing and precipitation, as discussed throughout section 4).
   Since basal ice was relatively enriched in all solutes, the addition of more basal ice would suggest the pond was increasingly *depleted* in these solutes.

6. Ln470: Are you excluding basal waters/melt from other potential sources to late season runoff? I again think several microorganisms detected in the pond waters might have originated and selected from subglacial sources too (as mentioned on the the first sentence of 4.4 Ln457 and the follow-up paragrap Ln 476-488). I suggest adding "subglacial waters" too to the sentence as potential source. But I agree with the overall interpretation of niche selection.
   Though it's not impossible that basal solutes contributed to late season runoff, based on the configuration of this drainage system (as described in Section 3.1 and 4.1), and isotopic composition of the water/ice (Section 3.2), we assume most originated from snow/supraglacial

melt and that basal contribution to be negligible.  We have clarified this interpretation in Section 4.1:

"Most of the melt generated on the Sverdrup Glacier drains ice-marginally and both the water isotopes and geochemistry of channel ice samples indicate snow and glacier ice as predominant water sources. We have no evidence to suggest that late season runoff draining into this subglacial tunnel originated from an upstream subglacial drainage network, but recognize subglacial water could contribute to late season water, for example, from melting basal ice along the glacier margin, or subglacial drainage from upstream"

We also amended line 476 to say "The meltwaters that drained into the subglacial system towards the end of the melt season originated from cryoconite holes, supraglacial streams, englacial ice, precipitation, and extraglacial aquatic or terrestrial sources, and potentially upstream subglacial sources (e.g. basal ice melt from ice marginal cliffs or subglacial water drainage)."

7. The above comment also makes me caution the conclusion statement that the detected bacterial populations most likely are generalists from supraglacial environments. To me they seem like typical subglacial populations; perhaps generalists yes and ultimately originated from extraglacial systems prior to glacial inception but I'm not sure this is what the authors are claiming (I still think the results are pretty cool though!).

You make a good point. Our argument that these organisms originated from the supraglacial environment is based on our assumption that late-season runoff is primarily of supraglacial/extraglacial origin. We feel we have provided strong evidence this is likely the case (section 4.1), though, in response to the previous comment, acknowledge it's not impossible that there is an upstream subglacial water/solute/microbe contribution. If present, this subglacial source could have introduced the gammaproteobacteria to the late season runoff, though still seems unlikely that small contributions of subglacial water would have yielded the high relative abundances of these OTUs in channel ice, since basal ice did not contain high relative abundances of these OTUs. Nevertheless, we agree it's worth acknowledging this potential source and the edits we made in response to Comment 6 do that.

8. There seems to be no discussion on why the cell abundance in the ponded water appears to be depleted relative to model. This seems a bit strange to me. Again, perhaps a higher contribution of basal ice to the model might help explain this (can see technical comments below)?

We expect that the cold temperatures, low nutrient content, absence of light, and freezing process would kill a portion of the microbes in the pond over the winter, resulting in a depleted reservoir of cells compared to the model. This would especially be the case if a large portion of the cells contained in late season runoff originated from the supraglacial or extraglacial environment and were not adapted to subglacial conditions.

Basal ice contained relatively high cell concentrations, so the addition of more subglacial sources would have yielded an even more depleted reservoir of cells in the pond. We have included explanation of the low cell concentrations in section 4.4 (line 480):

*"The shift in environmental conditions to those in the cold, dark, oligotrophic subglacial pond may have decimated populations not capable of survival in this system, selecting for generalist organisms that are better adapted to those conditions (Xu et al., 2021), and yielding lower cell concentrations in the pond than the model predicted (Figure 5a)."*

Data availability:

1. I believe a bit more information is needed regarding the geochemical models used (even if relatively simple). I.e. where/how were they run? Custom scripts? What program (Matlab, R, Exce?). If code was used, it should ideally be uploaded in a repository or included as supp material.

   All scripts were custom and programmed in Matlab 2020 using the equations in S1 and S2 – this detail was added to S1 and S2 and to the Methods section (Line 203, 226, 240).

2. Similar comment regarding the bioinformatics: e.g. might it be possible to upload the mothur logfile or command-line summary (e.g. batch file) to a repository or as part of Supp Mat.

   The workflow for the bioinformatics followed the standard operating procedure, available from the Mothur website and published by Kozich et al., 2013. The few user-defined settings are specified in the methods and the code is now available on Zenodo at 10.5281/zenodo.7942918.

3. I'm not sure the Zeonodo link for geochem data is correct and could not access the data (maybe my bad though).
   Zenodo link was set to restricted access, so that reviewers could anonymously request access (or at least this was the intent!). Nevertheless, it's now made public.

**Technical comments**

Ln 55: should also include Gill-Olivas 2023: https://bg.copernicus.org/articles/20/929/2023/

Thank you, this reference is now included.

Methods: consider changing the subscript acronyms in equations to capital letter to avoid confusion with lowercase roman numerals used for "steps" (e.g. Xii(i) for "incremental ice" could become XII(i) etc)

Revised as suggested

Methods: consider specifying that the geochem model was also applied to cell concentrations

Good point. We've renamed this as 'biogeochemical model' throughout so that it more intuitively includes parameters such as DOM and cell concentrations, though we've also revised the text in line 242 to explicitly state this: "We therefore used the same theory and equations as the $Cl^-$ model to produce a biogeochemical model (using custom scripts in Matlab 2020b) that simulates the concentration of other solutes and impurities (e.g. cells) in residual water and incremental ice…"

legend of Fig. 3a: isn't inversed? Here shows that red = 100% water and dark-blue = 100% frozen seems opposite to data and theoretical values? (also consider changing rainbow scale to alternative more colour-blind friendly palette (e.g. Viridis))

No, the legend is not inversed. Water (yellow) is isotopically heavy while last formed ice (blue) is isotopically light. Color bar has been changed to Viridis.

Fig. 5b: Should x-axis be "relative abundance to channel ice"?

Yes. Revised accordingly.

Ln 529: not sure the Zenodo link is correct?

Zenodo link was set to restricted access, so that reviewers could anonymously request access (or at least this was the intent!). Nevertheless, it's now made public.

Fig. S3a : I think bed and ice-surface are swapped in the legend. Also what are the capital "A" and "B" above both axes?

You're right, ice-surface and bed were swapped in the legend – this is now fixed. "A" and "B" refer to the cross section indicated in Figure 1. The caption for this figure has been revised to note that.